# Tinnitus Is Marginally Associated with Body Mass Index, Heart Rate and Arterial Blood Pressure: Results from a Large Clinical Sample

**DOI:** 10.3390/jcm12093321

**Published:** 2023-05-06

**Authors:** Berthold Langguth, Jan Bulla, Beate Fischer, Hansjoerg Baurecht, Martin Schecklmann, Steven C. Marcrum, Veronika Vielsmeier

**Affiliations:** 1Department of Psychiatry and Psychotherapy, Bezirksklinikum, University of Regensburg, Universitätsstraße 84, 93053 Regensburg, Germany; 2Department of Mathematics, University of Bergen, 5020 Bergen, Norway; 3Department of Epidemiology and Preventive Medicine, University of Regensburg, 93053 Regensburg, Germany; 4Department of Otorhinolaryngology, University of Regensburg, 93053 Regensburg, Germany

**Keywords:** tinnitus, cardiovascular, metabolic, risk factor, arterial hypertension, vegetative, sympathic, parasympathic

## Abstract

Introduction: This study aimed to explore whether body mass index (BMI), systolic blood pressure (RR syst), diastolic blood pressure (RR diast) or heart rate (HR) are associated with tinnitus status and/or severity. Methods: To that end, we evaluated the influence of tinnitus status and Tinnitus Handicap Inventory (THI) score on BMI, RR syst, RR diast and HR by comparing data from a large sample of patients presenting to a specialized tertiary referral clinic (N = 1066) with data from a population-based control group (N = 9885) by means of linear models. Results: Tinnitus patients had a significantly lower BMI and higher RR syst, RR diast and HR than non-tinnitus patients; however, the contribution of the case–control status to *R*^2^ was very small (0.1%, 0.7%, 1.4% and 0.4%, respectively). BMI had little predictive power for the THI score (higher BMI scores were related to higher THI scores; *R*^2^ = 0.5%) and neither RR syst, RR diast, nor HR showed a statistically significant association with THI. Discussion: Our findings suggest that HR, RR and BMI are at most marginally associated with tinnitus status and severity.

## 1. Introduction

Tinnitus, a common condition among adults with a global prevalence of about 15% [1], is typically defined as the perception of a sound without a corresponding external sound source [2]. Most people who experience tinnitus are not severely bothered by it; however, about 2.5% of the population will be substantially impaired by tinnitus in terms of quality of life [1]. In light of the highly variable impact of tinnitus, it was recently suggested to differentiate between *tinnitus* and *tinnitus disorder*. According to this proposal, *tinnitus* should be used to describe the auditory or sensory component, whereas *tinnitus disorder* reflects the auditory component and associated suffering, which can involve emotional distress, cognitive dysfunction, autonomic arousal, behavioral changes and functional disability [2].

Currently, the risk factors for tinnitus generation and maintenance are only incompletely understood [3,4]. Identifying and quantifying risk factors for tinnitus and tinnitus disorder is critical, as it could advance our understanding of the underlying pathophysiology, inform preventive interventions and support the formation of patient-centered management strategies.

In a recent systematic review, hearing loss, diabetes, depression and temporomandibular joint disorder were identified as statistically significant risk factors for developing tinnitus [4]. However, most available data about risk factors come from population-based epidemiological studies, which do not differentiate between *tinnitus* (the phantom perception of sound) and *tinnitus disorder* (tinnitus plus the associated suffering). Thus, risk factors for bothersome tinnitus or risk factors predictive of the severity of tinnitus suffering remain undetected through such studies. 

Relatively few studies have focused on clinical samples, which typically contain a higher proportion of patients with bothersome tinnitus compared to population-based studies. Data from such studies suggest a relationship between tinnitus and arterial hypertension [5,6], as well as between tinnitus and Body Mass Index (BMI) [6]. The notion of an association between tinnitus and arterial hypertension is also supported by a meta-analysis [7], which revealed the odds of experiencing tinnitus to be significantly increased (pooled odds ratio = 1.37) for persons with arterial hypertension. These findings fit with the theory that increased sympathetic activation is related to tinnitus distress [8]. An alternative explanation for increased arterial hypertension in tinnitus patients may be provided by findings of increased intima media thickness and arterial stiffness in tinnitus patients [9,10].

The primary aim of this study was to investigate the association of tinnitus status and tinnitus severity with arterial blood pressure (RR), HR and BMI. To this end, we analyzed data from a large sample of patients presenting to a specialized tertiary referral clinic and compared the findings with those from an age- and gender-matched control group [German National Cohort (NAKO)] [11]. Specifically, we were interested in whether the tinnitus patients and the control group differed in terms of RR, HR and BMI and whether there was an association between these factors and tinnitus severity, as assessed using a standardized questionnaire.

## 2. Methods

### 2.1. Clinical Sample

All patients presenting with chronic subjective tinnitus to the Interdisciplinary Tinnitus Center at the University of Regensburg (Regensburg, Germany, a tertiary referral center) from 1 April 2015 to 30 April 2021 were invited to participate in the study. A total of 1066 Patients (381 females, 685 males) provided informed consent that data could be gathered and analyzed for the Tinnitus Research Initiative Database [12], which was approved by the Ethics Committee of the University Hospital of Regensburg (Germany; reference number 08/046). Demographic and clinical data of the clinical patient sample is provided in Table 1.

### 2.2. Control Sample

The control group consisted of 9885 participants (4922 females, 4963 males) aged 20–75 years from the population-based German National Cohort (NAKO). All study participants were identified based on an age and sex-stratified sample randomly drawn from compulsory registries of residents within the study area of Regensburg.

All persons within the control group completed a recruitment protocol, which included provision of detailed information on the study procedures, informed consent collection, completion of a standardized, computer-assisted personal interview (including whether the person had ever been diagnosed with tinnitus by a physician), and basic physical and medical examinations (e.g., height, weight, HR and blood pressure). The study design, sampling method and data collection have been described in detail in previous work [13]. Demographic and clinical data of the NAKO control-group sample are provided in Table 2.

### 2.3. Statistical Methods

All statistical analyses were carried out using the statistical software R (R Core Team, 2021), version 4.0.5. For our main analyses, we assessed the potential impact of the presence of tinnitus, which was captured based on the case–control status, on the response variables BMI, systolic blood pressure, diastolic blood pressure, and heart rate by means of linear models. The NAKO control group was modeled as the reference level. To determine the best model for each of the response variables, we considered including age and sex as covariates in our linear models. For the sex factor variable, the reference level is “female”.

We followed a bottom-up approach to determine the best model and assessed potential models by both Akaike information criterion (AIC) and Bayesian information criterion (BIC). Starting with the simplest model, we successively included covariates and interactions with case–control-status most strongly associated with the outcome measured in terms of R^2^. Thus, we proceeded in decreasing order of impact (in terms of R^2^). In addition, F-tests served as model comparison checks. Finally, we checked for normality of the residuals of the final model using QQ-plots and the Shapiro–Wilk test. When normality was not supported by either visual inspection or a statistical test, we repeated the analysis with a robust MM-type regression approach. For this regression type, we relied on the lmrob function from the robustbase package [14], with the estimation setting KS2014, as suggested by Koller & Stahel [15]. Consistency checks of our regression models were carried out via simple χ^2^-tests for independence. The general level of significance in this study is *p* < 0.05.

## 3. Results

### 3.1. Comparison with the Population-Based Control Group

All of the best robust linear regression models identified in this study incorporated the group effect in conjunction with the covariates age and sex, with total model fits in terms of *R^2^* ranging from 3.9% for heart rate to 21.5% for syst. RR (Table 3). However, the contribution of the group variable to the observed *R^2^* was very small (0.1%, 0.7%, 1.4% and 0.4% on the models for BMI, syst. RR, diast. RR and heart rate, respectively).

Analysis revealed a correlation between case–control status and hypertension yes/no (defined by syst RR ≥ 40mmHg), as independence of the group membership and dichotomized syst. RR (<140: “low”, ≥140: “high”) was rejected by a simple χ^2^-test (χ^2^ = 109.01, *p*-value < 0.001). The same applied to a multi-categorical subdivision of syst. RR values: Changing from dichotomization to polychotomization with interval borders 120, 130, 140, 160, and 180 (representing the categories “optimal”, “normal”, “still normal”, “hypertonic degree I”, “hypertonic degree II”, and “hypertonic degree III”) gave a similar result (χ^2^ = 169.99, *p*-value < 0.001). Furthermore, we saw a correlation with dichotomized HR (<90: “low”, ≥90: “high”; independence of group membership and HR rejected: χ^2^ = 218.44, *p*-value < 0.001).

### 3.2. Tinnitus Handicap Inventory Analysis within the Clinical Patient Sample

In addition to the between-group comparisons presented above, we also analyzed whether tinnitus distress assessed by the THI can be predicted by any of the variables collected for the clinical sample. Table 4 presents the “best” model in the sense that BMI could be identified as a significant predictor. However, this model has a total *R^2^* of only 1.1% (BMI contribution of approximately 0.43%), suggesting very limited predictive power. When including RR syst, RR diast, or HR as alternative predictors, no statistically significant results were obtained (results not shown).

## 4. Discussion

The aim of this study was to investigate the relationship between tinnitus, HR, RR, and BMI using a large clinical sample and a control group from a population-based cohort. The main findings of our study were that tinnitus patients had a lower BMI than controls, but a higher heart rate and higher values for systolic and diastolic blood pressure. While statistically significant, observed effects were small. Specifically, the group factor explained a maximum of 1.4% of the total variance. Thus, the size of the observed effects does not achieve clinical relevance.

The relationship between RR and tinnitus has been investigated in a relatively large number of studies, whereas less work is available describing the relationship between tinnitus, BMI, and HR. Further, even within available studies, the pattern of results is unclear. For example, while Mahboubi et al. reported no relationship between BMI and tinnitus in adolescents [16], a population-based study from Korea suggested both a low and a high BMI to be associated with tinnitus in premenopausal women [17]. In a small clinical sample incorporating 46 participants with tinnitus and 74 controls, a high BMI in men was reported as being a significant predictor for tinnitus [6]. This finding has found support in recent work utilizing the extensive UK biobank dataset, which also revealed an association between tinnitus and a higher BMI [18]. In our clinical sample, BMI was slightly reduced relative to controls. Further, higher tinnitus severity was associated with a higher BMI in this group, though the correlation was weak. The observed variability in the relationship between tinnitus and BMI might be due to differences in sample selection and eventually also due to variability in tinnitus definition [2].

Our data provide only limited support for the notion that tinnitus distress is related to increased sympathetic activation [8,19]. While we found a slightly increased heart rate in tinnitus patients, no relationship between heart rate and tinnitus handicap could be identified. The relationship of tinnitus distress and sympathetic activation has already been questioned by an experimental approach in which stress reactivity was tested in small samples of tinnitus patients and controls. This study showed decreased heart rate not only during relaxation, but also during a stressful tinnitus-related event [20] highlighting the complexity of the association of tinnitus (distress) with indicators of sympathetic activation.

A large number of studies have investigated the relationship between blood pressure and tinnitus. Both studies utilizing clinical samples [5,6] and population-based studies [18,21,22,23,24,25] have suggested arterial hypertension to be associated with tinnitus and tinnitus handicap [26]. However, other studies could not confirm such an association [16,27,28]. In our study, we found a slightly increased RR syst and RR diast in patients with tinnitus, but no correlation with tinnitus handicap.

Taken together, our findings suggest only very weak associations between HR, RR, BMI and tinnitus. Interestingly, tinnitus handicap, as measured with the THI, showed no association with HR and RR and only a very weak association with BMI. We initially hypothesized that HR, RR and BMI would be more strongly related to tinnitus disability. Further, we anticipated that people with low tinnitus disability and people with high tinnitus disability, or according to recent nomenclature, people with tinnitus and people with tinnitus disorder [2], would significantly differ in terms of HR, RR or BMI. Present results suggest that tinnitus severity has only a minor impact on these relationships. Indeed, the general weakness of the association between tinnitus, BMI, RR and HR in our clinical sample closely mirrored that for a large control dataset and is broadly consistent with results of recent meta-analyses [4,7].

### Limitations

Tinnitus heterogeneity was not considered in our analysis. Subtyping of tinnitus patients according to pulsatile character or tinnitus pitch might reveal different results and should be considered in future research. While RR, HR and BMI measurements were carried out in both the clinical sample and the population-based sample in a standardized way, the protocol was not identical. Different instruments were used for measurement and the order of measurements, as well as the time intervals between measurements differed. While an effect of these methodological differences on study results cannot be entirely excluded, they were minor and are not believed to have impacted results to a meaningful extent.

## Figures and Tables

**Table 1 jcm-12-03321-t001:** Basic descriptive statistics of the clinical patient sample.

	Female (N = 381)	Male (N = 685)	Total (N = 1066)
Age [yrs]	51.1 (±13.7)	51.4 (±12.2)	51.3 (±12.8)
THI score	49.4 (±22.2)	47.3 (±23.1)	48.0 (±22.8)
Missing THI score	5 (1.3%)	2 (0.3%)	7 (0.7%)
Height [m]	1.66 (±0.069)	1.78 (±0.069)	1.74 (±0.091)
Weight [kg]	68.7 (±12.8)	86.8 (±15.0)	80.3 (±16.7)
BMI [kg/m^2^]	24.9 (±4.5)	27.2 (±4.3)	26.4 (±4.5)
HR [bpm]	71.5 (±10.6)	69.1 (±11.6)	70.0 (±11.3)
RR syst [mmHg]	132 (±19.8)	139 (±17.2)	136 (±18.5)
RR diast [mmHg]	82.4 (±11.9)	86.1 (±11.7)	84.8 (±11.9)

THI. Tinnitus Handicap Inventory, BMI: Body Mass index, HR: Heart rate; RR syst: systolic blood pressure; RR diast: diastolic blood pressure.

**Table 2 jcm-12-03321-t002:** Basic descriptive statistics of the NAKO control group sample.

	Female (N = 4922)	Male (N = 4963)	Total (N = 9885)
Age [yrs]	50.0 (±12.7)	49.9 (±12.8)	50.0 (±12.8)
Height [m]	1.65 (±0.065)	178 (±0.070)	171 (±0.095)
Weight [kg]	70.9 (±14.9)	87.0 (±15.1)	79.0 (±17.1)
BMI [kg/m^2^]	26.2 (±5.5)	27.4 (±4.5)	26.8 (±5.1)
HR [bpm]	68.8 (±10.2)	67.3 (±11.4)	68.0 (±10.8)
RR syst [mmHg]	126 (±17.8)	134 (±15.8)	130 (±17.2)
RR diast [mmHg]	78.1 (±9.9)	81.7 (±10.1)	79.9 (±10.2)

BMI: Body Mass index, HR: Heart rate; RR syst: systolic blood pressure; RR diast: diastolic blood pressure.

**Table 3 jcm-12-03321-t003:** Results of a robust regression model predicting BMI with group effect and covariates age and gender (top panel). The three following panels show the corresponding models for RR syst, RR diast, and HR, respectively, with group effect and covariates age, gender, and BMI. Reference level for the group factor variable is the NAKO cohort. For the gender factor variable, the reference level is “female”.

Term	Estimate	Std. Error	*t*-Statistic	*p*-Value
Body Mass Index [kg/m^2^] *R^2^* = 11.7%				
(Intercept)	20.34	0.17	120.47	<0.001
Group	−0.57	0.14	−4.16	<0.001
Age	0.10	0.00	31.99	<0.001
Gender	1.69	0.08	20.69	<0.001
Systolic blood pressure [mmHg] *R^2^* = 21.5%				
(Intercept)	91.05	0.87	105.15	<0.001
Group	4.91	0.49	10.03	<0.001
Age	0.39	0.01	33.46	<0.001
Gender	7.42	0.29	25.49	<0.001
BMI	0.56	0.03	18.64	<0.001
Diastolic blood pressure [mmHg] *R^2^* = 13.5%				
(Intercept)	60.40	0.56	108.38	<0.001
Group	4.13	0.32	13.05	<0.001
Age	0.13	0.01	17.68	<0.001
Gender	3.11	0.19	16.67	<0.001
BMI	0.41	0.02	21.44	<0.001
Heart rate [bpm] *R^2^* = 3.9%				
(Intercept)	62.11	0.61	102.15	<0.001
Group	2.30	0.34	6.77	<0.001
Age	−0.06	0.01	−7.48	<0.001
Gender	−2.43	0.20	−11.97	<0.001
BMI	0.36	0.02	17.34	<0.001

**Table 4 jcm-12-03321-t004:** Results of a robust regression model predicting THI with group effect and covariates age, gender, and BMI.

Term	Estimate	Std. Error	t-Statistic	*p*-Value
THI score R^2^ = 1.1%				
(Intercept)	35.17	4.89	7.18	<0.001
Age	0.06	0.06	1.03	0.304
Sex	−3.39	1.61	−2.11	0.035
BMI	0.43	0.17	2.47	0.014

## Data Availability

Data is unavailable due to privacy and ethical restrictions (lack of informed consent on data sharing).

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
