# Peer review of "Tinnitus Is Marginally Associated with Body Mass Index, Heart Rate and Arterial Blood Pressure: Results from a Large Clinical Sample"

_jcm, 2023, doi:10.3390/jcm12093321_

Round 1

Reviewer 1 Report

This study investigates the association between body mass index (BMI), systolic and diastolic blood pressure, heart rate (HR) and tinnitus. The study assesses differences between tinnitus and controls as well as the association between these factors and tinnitus severity as measured by THI. A large population of tinnitus and controls is used, and factors age and sex are used as covariates in the analysis. The analysis shows differences in BMI, HR and blood pressure with small observed effects that the authors describe as not clinically relevant.

While the findings are well described, there are quite variable findings in terms of the factors assessed, in literature. Other than sample size, what else do the authors think contributes to these variable findings (in particular lines 158-165)?

A few other clarifications:

-Line 58:  The primary aim of this study was to investigate the influence of tinnitus status and  tinnitus severity on arterial blood pressure, heart rate (HR) and BMI”- Is  the study aim to assess the ‘influence of tinnitus’ on these factors or their association? I believe the latter as the direction of the effect isn’t known (e.g. tinnitus causing high BP or the reverse).  

-The size of the clinical sample can be mentioned under 2.1 (similar to the first sentence under 2.2).

-A few items are mentioned in the Discussion with no clear reference in the background. These are mainly, sympathetic activation and the hypothesis (“We initially hypothesized that HR, RR and BMI would be more strongly related to tinnitus disability”). Some background on sympathetic activation should be added to the Introduction, since it is an important basis of this study. The hypothesis would also be better placed in the Introduction.

-Line 194 states the protocols in the two groups were not identical. Please add a sentence or two on what the main differences were.  

Author Response

This study investigates the association between body mass index (BMI), systolic and diastolic blood pressure, heart rate (HR) and tinnitus. The study assesses differences between tinnitus and controls as well as the association between these factors and tinnitus severity as measured by THI. A large population of tinnitus and controls is used, and factors age and sex are used as covariates in the analysis. The analysis shows differences in BMI, HR and blood pressure with small observed effects that the authors describe as not clinically relevant.

While the findings are well described, there are quite variable findings in terms of the factors assessed, in literature. Other than sample size, what else do the authors think contributes to these variable findings (in particular lines 158-165)?

Answer:

We added a sentence in which we discuss the reasons for the variable finding:

 The observed variability in the relationship between tinnitus and BMI might be due to differences in sample selection and eventually also due to variability in tinnitus definition (2).

A few other clarifications:

-Line 58:  “The primary aim of this study was to investigate the influence of tinnitus status and  tinnitus severity on arterial blood pressure, heart rate (HR) and BMI”- Is  the study aim to assess the ‘influence of tinnitus’ on these factors or their association? I believe the latter as the direction of the effect isn’t known (e.g. tinnitus causing high BP or the reverse). 

Answer:

We agree with the reviewer, that the direction of the effect is not clear and therefore replaced “influence” by “association”

-The size of the clinical sample can be mentioned under 2.1 (similar to the first sentence under 2.2).

Answer:

We added the sample size as proposed by the reviewer

-A few items are mentioned in the Discussion with no clear reference in the background. These are mainly, sympathetic activation and the hypothesis (“We initially hypothesized that HR, RR and BMI would be more strongly related to tinnitus disability”). Some background on sympathetic activation should be added to the Introduction, since it is an important basis of this study. The hypothesis would also be better placed in the Introduction.

Answer:

We agree with the reviewer and added the following sentence in the introduction:

These findings fit with the theory, that increased sympathetic activation is related to tinnitus distress (8).

-Line 194 states the protocols in the two groups were not identical. Please add a sentence or two on what the main differences were.  

Answer:

We added a sentence with the main differences:

 Different instruments were used for measurement and the order of measurements as well as the time intervals between measurements differed.

Reviewer 2 Report

The manuscript presents an important issue of  the tinnitus status and severity on Arterial Blood Pressure, Heart Rate (HR) and BMI. The major limitation concerns a protocol not identical standardized way of RR, HR and BMI carried out measurements in both the clinical and the population-based sample.

Tinnitus status and severity might also be associated with arterial  Intima Media Thickness (IMT) as preclinical atherosclerotic biomarker and other biomarkers of hemodynamic instability (ANP) in order to better describe this clinical model.

Author Response

The manuscript presents an important issue of  the tinnitus status and severity on Arterial Blood Pressure, Heart Rate (HR) and BMI. The major limitation concerns a protocol not identical standardized way of RR, HR and BMI carried out measurements in both the clinical and the population-based sample.

Answer:

We provided more details about the protocol differences in the measurement of RR. HR and BMI:

Different instruments were used for measurement and the order of measurements as well as the time intervals between measurements differed.

Tinnitus status and severity might also be associated with arterial  Intima Media Thickness (IMT) as preclinical atherosclerotic biomarker and other biomarkers of hemodynamic instability (ANP) in order to better describe this clinical model.

Answer:

We want to thank the reviewer for this comment and added the following sentence in the introduction:

An alternative explanation for increased arterial hypertension in tinnitus patients may be provided by findings of increased intima media thickness and arterial stiffness in tinnitus patients (9, 10)

Reviewer 3 Report

Title: Tinnitus is marginally associated with body mass index, heart 2 rate and arterial blood pressure: Results from a large clinical 3 sample

1) In my opinion, this paper on tinnitus lacks research on the type of tinnitus felt by people. Relating tinnitus with circulatory problems, it would be very interesting to know what the pitch of tinnitus is, taking into account that a pitch of low frequencies is more related to malfunction of the temporomandibular joint, for example, or an acute pitch, more related to problems of the inner ear or central nervous system. Cardiovascular problems usually lead to pulsatile tinnitus. This research was not done in this work, but it would be very interesting to realize this relationship between heart rate and pulsatile tinnitus.

2) For a better visualization of the results, the table 3 should be expressed by graphs, besides that it would make the paper more appealing and friendly to interpret.

3)  Abbreviations:

Throughout the text check and correct the abbreviations: sys RR, diast RR and HR, RR syst, RR diast (these in the Abstract) and sometimes in the text appears HR and then in full “heart rate”. Verify, for example: Lines 97, 98, 119, 121, 124, 128, 129, 130, 143, 148, 151, 155, 168, 169, 179, table 1 and 2.

4) Line 29: … prevalence of about 15%...

Where?

I hope I have contributed to better paper, or who knows, future works.

Author Response

1) In my opinion, this paper on tinnitus lacks research on the type of tinnitus felt by people. Relating tinnitus with circulatory problems, it would be very interesting to know what the pitch of tinnitus is, taking into account that a pitch of low frequencies is more related to malfunction of the temporomandibular joint, for example, or an acute pitch, more related to problems of the inner ear or central nervous system. Cardiovascular problems usually lead to pulsatile tinnitus. This research was not done in this work, but it would be very interesting to realize this relationship between heart rate and pulsatile tinnitus.

 Answer:

We agree with the reviewer, that subtypisation according to pulsatile character or tinnitus pitch might reveal different results. We added this important point in the limitation section:

Tinnitus heterogeneity was not considered in our analysis. Subtypisation of tinnitus patients according to pulsatile character or tinnitus pitch might reveal different results and should be considered in future research.

2) For a better visualization of the results, the table 3 should be expressed by graphs, besides that it would make the paper more appealing and friendly to interpret.

Answer:

We agree with the reviewer, that a graphical illustration of the results would be desireable. However, unfortunately the results of the regression model with covariates cannot be cannot be well depicted graphically

3)  Abbreviations:

Throughout the text check and correct the abbreviations: sys RR, diast RR and HR, RR syst, RR diast (these in the Abstract) and sometimes in the text appears HR and then in full “heart rate”. Verify, for example: Lines 97, 98, 119, 121, 124, 128, 129, 130, 143, 148, 151, 155, 168, 169, 179, table 1 and 2.

 Answer:

We harmonized the abbreviations throughout abstract and text

4) Line 29: … prevalence of about 15%...

Where?

 Answer:

The number comes from a review reporting the global prevalence. We added “global”

I hope I have contributed to better paper, or who knows, future works.

Answer:

Thank you for your contribution
